# What Would Miffy Do? Applying Informed Consent by Proxy to All Sentient Animals

**DOI:** 10.3390/ani14182643

**Published:** 2024-09-12

**Authors:** Monique R. E. Janssens

**Affiliations:** Ethisch Bedrijf, 3011 XL Rotterdam, The Netherlands; moniquejanssens@ethischbedrijf.nl

**Keywords:** consent, dissent, assent, animal experiments, ethics, representation, by proxy

## Abstract

**Simple Summary:**

To respect the interests of sentient animals, we should consider asking for their consent before using them for human purposes. While mentally competent humans can provide informed consent by understanding the risks and benefits, non-human animals cannot fully understand these aspects. Thus, we need other methods to determine their preferences and choices. A promising approach is to obtain informed consent by proxy from humans, alongside seeking assent from the animals where possible and paying attention to any signs of dissent.

**Abstract:**

If we want to take sentient non-human animals and their interests seriously, we can try to ask for their consent before using them for human purposes. With mentally competent humans, we speak of informed consent: for them to participate in scientific studies, for example, it is required that they consent explicitly, in full understanding of the risks and benefits. This full understanding cannot be expected from non-human animals. We must therefore look for ways to know what they want and to estimate what they would do if they had a deep understanding of their options and the consequences of these options for themselves and others. This concept is explored by applying it to animal experiments. The most promising method is to gain informed consent by proxy from thoroughly informed competent humans, in combination with seeking assent where possible and being alert to dissent.

## 1. Introduction

When I give lectures in animal ethics, I always bring my Miffy keyring as a mascot. Miffy—originally Nijntje—is a rabbit cartoon character from the children’s books of Dutch author–illustrator Dick Bruna. To me the small rabbit is a symbol of how differently humans treat live non-human animals in different contexts: in households; in breed shows; in research; in the food industry; or in the wild, as part of nature, a pest, or a hunting target.

In our human-dominated society, we often do as we please with non-human animals, and we hardly ever ask for their consent. We agree that one cannot do that with humans, especially since the broad acceptance of Kantian philosophy and specifically the discovery that the Nazis had experimented on humans against their will. We cannot use the humanity of others merely as a means to achieve certain goals. One should also consider the interests and the dignity of that human as a person [1].

If we do want to use humans for useful purposes, we ask for their consent, directly or indirectly. If we need help, we ask our friends or family, who can choose to accept or refuse. The person who delivers our pizza has signed a contract in which they consent to work as a pizza delivery person and receive a salary in return. For participation in scientific research, informed consent is required by law, meaning that potential participants are informed in an understandable way about the treatment and the possible consequences and sign the information to confirm that they participate voluntarily. We can take the voluntariness of their participation seriously if they are competent, adequately informed, and not coerced. Being adequately informed also means that they are able to comprehend the nature and purpose of the experiment and their participation; the procedures of the research project; and the burdens, risks, and benefits for themselves and for others, including those of alternative treatments; they also must be able to communicate their choice [2,3]. Informed consent usually takes the form of paperwork combined with a face-to-face interview where questions can be asked and unclear elements can be clarified [4].

As the scientific knowledge of non-human animal cognition and emotions expands, the role of animals in testing is under revision [5]. Since the political turn in animal ethics, political scientists have argued that animals are agents who can very well make decisions about their needs and preferences and communicate these [6]. The ways by which they can do this vary with species and individuals. Basically, this is not so different from the way it works with humans, who come in a broad variety of capacities as well, depending on age, education, social situation, experience, intelligence, talents, abilities, et cetera.

Now and then, living beings, human or non-human, need other living beings for achieving specific purposes. Without the possibility of involving others, many useful activities leading to pleasurable lives would become impossible. We need parents or other guardians to grow up safely. We need teachers to educate us. We need friends to have social relationships or to throw a party. We need the police to keep our environment safe. But using others without their consent leads to horrifying forms of suffering: confinement, slavery, oppression, physical and sexual violation, and slaughter. Therefore, we should find ways of living with other human and non-human animals, listening to them, talking with them, asking for their consent in situations where we are in power, and abstaining from the activities they do not consent with.

The aim of this paper is to explore which of the concepts of assent, dissent, and consent is the most helpful in cases of the use of sentient non-human animals for human purposes. After having concluded that informed consent by proxy is the most helpful one, we will explore how a form of informed consent can be obtained from sentient animals. It is important to find ways to do this, because of the large-scale use of non-human animals.

In this paper, I take as a premise that sentient animals have moral standing and are active subjects of their own lives, based on the work of philosophers of several ethical approaches, such as utilitarianism [7], deontology [8,9], and the virtue-based capabilities approach [10].

## 2. Materials and Methods

The approach of this paper is philosophical and normative: we will review the key publications on assent, dissent, and consent and see what they can contribute to our case. We will find a preliminary solution and apply that to a few exemplary cases.

## 3. Results

### 3.1. Taking Animals Seriously: Assent, Dissent, and Representation

Sentient non-human animals can participate in moral responsibility practices as moral addressors who communicate whether they appreciate or disapprove of specific behaviour towards them [11]. Think for example of a dog that growls because it does not want to be approached. But does this hold for more abstract situations or in advance of the behaviour? This raises communicative problems. Informed consent in particular can be problematic with those who do not understand all the ins and outs of the language the other uses or who cannot “understand, form, and communicate complex intentions about normative concepts like rights and duties” [12]. This is even the case for illiterate humans or when the partners in communication use different human languages (including, e.g., sign language).

Most decisions we make lead to positive or negative impacts for both ourselves and others. Therefore, many choices are moral choices. Can sentient non-human animals make moral choices about themselves and others and act consciously upon these choices? Not a lot is known about the moral agency of animals, which one could define as being able to (morally) understand, reflect on, and evaluate potential or actual actions, omissions, or character traits of oneself and others [11]. Nevertheless, we do know that in some species altruistic actions have been observed, such as bonobos and chimpanzees offering consolation to each other [13], rats helping their co-rats to treats without any traceable advantage for themselves [14], or mice comforting other mice that were exposed to pain or stress [15]. The problem that informed consent cannot be obtained from most non-human animals in most cases holds for both choices about themselves and choices about themselves and others.

For both humans and non-humans, several concepts have evolved to deal with this problem: assent (an active willingness to collaborate without being fully informed or being able to fully understand the implications of collaboration), alertness to dissent (crying, struggling, and refusing to move), and consent. We will now briefly discuss each of them.

### 3.2. Assent

Assent is an indication that the subject is prepared to participate in an activity, based on some information, or “wilful affirmation [that] requires observable behaviour that gives us reason to believe that an individual desires, prefers, or chooses the option or state of affairs” [12]. For example, subjects are shown what will happen when they embark on an activity. How profound that information is and what the expected level of comprehension should be remains unclear [16]. The aforementioned lack of shared language can make the communication insufficient [17].

In animal experimentation, assent may be applicable to repeated tests with experienced primates, who will at a certain point know what to expect (e.g., a fun game, followed by compliments and cuddles; or a short moment of pain, followed by treats and compliments), but it seems too unreliable for application to situations with animals who have less experience with the specific situation or with communicating with humans.

If animals are very experienced, you could say that they are more or less trained. Training animals is also conducted actively and consciously. It can be misleading though, because training tempts animals into agreeing to something that is not always in their best interests. On the other hand, it can lead to a relationship of trust between caretaker and animal, so that the animals will be more relaxed and, as a result, show more of their fears and preferences [18]. Furthermore, if the procedures will be executed and repeated anyway, training can reduce the stress and make the animals feel more comfortable, which reduces the burden for the animal [19]. This means it can truly lead to better welfare in a given situation. The downside is that it is exactly the trusting relationship that is created between animal and caretaker that makes exploitation and betrayal possible or at least easier [20]. This means that experience of animals, as well as lack of experience, is problematic for applying the concept of assent.

Another critical point is that there is no room for the moral agency of the non-human participant, because the implications for others are unknown to the animal. They can be explained, but the animal will probably not understand. In many cases, the implications will not appear immediately and will therefore be difficult to make clear.

### 3.3. Alertness to Dissent

Sometimes animals explicitly dissent: they can refuse to collaborate in specific activities by expressing their objection through sounds or other behaviour, like struggling, including staying still or hiding when movement is required [12,16,17]. Again, this phenomenon is mainly explored with laboratory animals. It has been argued that researchers who work with laboratory animals should be educated in assessing dissent from the animals they work with and stay on the safe side in case of doubt [17]. Responding to dissent in animal research has been mentioned as the only way to take laboratory animals seriously [16].

It is important to note though that Kantin and Wendler discuss dissent in the context of experiments with animal agents, defined as those who value their ability to play an active role in shaping their lives and themselves, for example, humans, great apes, and dolphins. They argue that researchers should solicit assent, and, if that is not possible, respect agents’ dissent. They also point out that being able to make choices and to live a life in peace (e.g., in the animal facility of the research institution) can compensate for small infringements on the preferences of these animal agents as shown by assent or dissent [17]. But even within these groups of acknowledged animal agents, moral issues can arise. It is conceivable that chimpanzees would dissent to the administration of anaesthesia or analgesia that might have been in their interest. Furthermore, one can also imagine them collaborating in simple behavioural tasks that yield treats, not knowing that the design of the experiment requires that they ultimately be killed to study their brain, without them getting the opportunity to refuse that final part of the experiment.

Another problem with the concepts of assent and dissent is that they cannot easily be broadened to other animal species, even if it is taken into account that many more species are nowadays seen as agents [6]. Not all mice will be able to predict what is going to happen, and not all activities are repeated so that the animals can build experience. Even more issues arise if we look at non-experimental practices. Many animals show thorough signs of dissent when they are taken to the vet, even if that is for treatment that will prevent disease, enhance their welfare, or even save their lives. At that point, they do not know it is in their interest to cooperate. Kantin and Wendler conclude as well that the absence of dissent (acquiescence) can be observed towards activities that are not in the interest of the animal [17]. Therefore, my objection to this approach is that non-human animal agents, be it by the definition of Kantin and Wendler or by a broader definition, will in many cases not be able to oversee the consequences of their choice to assent or dissent.

Finally, just like with assent, alertness to dissent cannot facilitate the moral agency of sentient non-human animals, if that capability is present. If an animal refuses a specific action or involvement, we cannot know what the animal would choose if it knew how it could help others by giving assent. My conclusion is that we should look for an approach that works better.

### 3.4. Consent

Applying the concept of consent has been suggested as well. As a normative power, consent releases the other from a duty. It is important though that it is exercised intentionally [12]. This is problematic as well, because intention requires understanding of the full situation and the consequences of different actions and non-actions, and in many cases non-human animals will not be able to fully grasp the situation.

What if we make use of informed consent by proxy, which is applied in research involving human subjects who are not competent to give informed consent, for example, young children or people with mental disorders or disabilities [16,21]? These participants are represented by personal consultees, legally authorized representatives, or authorized third parties [22], mostly family members who try their best to represent their loved ones. Nevertheless, the ability of proxies to represent others can be problematic. When family members were asked about life and death decisions—whether preceding treatment or not—researchers identified biases towards the proxy’s own views, values, or interests, and sometimes a hesitance to take responsibility for such heavy decisions [23]. In another study with elderly patients and their proxies, almost one-third of the proxies who believed that the elderly patient would refuse consent did give consent nevertheless [21].

The same issues could occur when consent by proxy is applied to non-human animals living with humans, maybe even in a stronger way because of the unequal power relations between humans and non-human animals [22]. When vets have to decide about the treatment of companion animals, breeding animals, or farmed animals, they ask their owners’ opinion [24]. The owners will try to assess to a lesser or greater extent what is important to the subjects (e.g., in which cases they will still enjoy life, in which cases they will want to end suffering, or what a life with limitations would mean to them). This will always be the owner’s biased interpretation of the animal’s preferences. Just like with humans, non-human animals bear the risk of misrepresentation [22]. The legal owners of non-human animals can have emotional and economic interests that can play a role on both sides of the coin. Emotional attachment can lead to continuing treatment beyond the rational or to premature killing when the owner cannot stand to witness any more suffering, even if that would be in the interest of the animal. (However, many veterinarians feel free to refuse to kill the animal if they do not agree [24].) Economic considerations can make an owner refuse expensive treatment that is in the interest of the animal or favour the longer treatment of an expensive purebred animal.

Would it then be better if the proxy, in this case the owner, had an independent position? We already argued that communicating with non-human animals is sometimes possible but often difficult, much like it is with young children or elderly people with dementia. People close to non-human animals may be able to do this in the best way, although their relationship can also stand in the way of independent judgement. Therefore, expert knowledge about the species is important as well. In the ideal world, to be the optimal assessor of the situation, the proxy would have both personal but independent experience with the animal and expert knowledge of the species, or a combination of proxies would be involved with each one of these characteristics.

Last but not least, informed consent by proxy acknowledges moral agency. It offers the proxy, and therefore the animal, the opportunity to look not only to their own interests but also to those of others. What remains problematic, though, is that the level of altruism differs not only between species but also among individuals. Just like humans, sentient non-human animals of the same species can vary in their level of altruism on an individual basis, depending on their character and their experiences. This observation calls for extra attention in choosing the right representative.

## 4. Discussion

### 4.1. Best Option: Informed Consent by Proxy

If we realise that in most cases it is impossible for a non-human animal to conceive of their full situation, including the consequences of their choices, and that even if they are competent agents, they can only make small, short-term choices, then informed consent by an expert acquaintance (or an expert and an acquaintance) as a proxy seems the best option to make sure the most ethical choices are made in complex situations. We are not talking about simple situations, such as letting a dog decide which path to take during a walk. And even in a case like this, their human guardian may have read information about a road block that the dog is not aware of. On the other hand, a dog may smell a distant fire that the human is not aware of, so let us not overestimate human knowledge. In the best case, the two will decide together. However, this type of decision can easily be dealt with in daily life. We will concentrate on more complex decision making.

In most complex cases, humans can know more about the options, chances, and consequences of actions than non-human animals. The way to move forward in these cases is to assess the interests of the animals as best we can. This means that the choice of the proxy or proxies is important: those who can understand the information and have this aforementioned mix of independent understanding of the individual and knowledge of the species. In addition to this, knowledge and insights can be enhanced through reading about the welfare and ethology of the species, consulting experts in these fields, or talking to the human guardian of the animal. In addition, one can observe the subject, listen to them, and communicate with them [6,25,26].

Representation is also advocated by Nussbaum [10] (p. 97):

“Animals do not speak human language, but they have a wide range of language-like ways of communicating about their situation (…), and if we humans happen to be in the driver’s seat politically, it should be our responsibility to attend to those voices, to figure out how animals are doing and what obstacles they face. We already do this for human beings who have disabilities that prevent them from participating in political life in the usual way: we give them guardians or “collaborators” to whom they express their situation and who become adept readers of their needs”.

Despite the risk of bias through the close relationship, Nussbaum does not hesitate to judge a situation on behalf of her companion cat. She offers an example of this line of thinking [10] (p. 216). To her own practical considerations about neutering the cat or not, which may be burdens such as aggression, spraying, and running away, she adds considerations on behalf of the cat. On the one hand, the cat would probably like to reproduce, be a parent, and take care of kittens. On the other hand, being pregnant several times in a row can be quite exhausting. More importantly, if the parent who is protective by nature knew that their offspring would face a grim future as redundant pets who may spend part of their lives in a shelter, they probably would not choose to have them. These are risks even the human involved can not easily foresee in their entirety, let alone a cat. Nussbaum concludes that she can imagine the cat’s hypothetical consent to be neutered. Still we find several uncertainties to this assessment. Some of the arguments attributed to the cat may hold stronger for females. Would a male cat consent as well? We humans have no idea what the sexual aspects of fertility feel like for a cat. Is the urge to mate a pleasure or a burden? This case is illuminating, but it also underlines the importance of the expertise of proxies and the limits to representation.

Nevertheless, Nussbaum advocates for the representation of non-human animals by humans in court and in policymaking. We could even try to broaden it to every important ethical decision involving non-human animals we as humans make. What if we consciously put ourselves in the animal’s place, based on all the expertise we have at hand, and imagined what the different options mean for the animal and what the animal would do? In other words, what if we always tried to answer the following question: what would Miffy do?

Critics might raise the objection that thinking and choosing on behalf of the animal tends towards anthropomorphism: ascribing human mental experiences to animals. However, the fear of anthropomorphism has been overstated. Just as there is the risk of anthropomorphism, there is also the risk of being in anthropodenial: underestimating the similarities between humans and non-human animals [27]. This means that the challenge lies in empathizing as a proxy with the animal in such a way that both the distinctive characteristics of the species and the comprehensible similarities to humans, as well as the individual animal’s personality, are properly represented.

### 4.2. Animal Experiments as a Case Study

To explore what this would imply, we will now look into animal experimentation as a case study. One would expect, prima facie, that those who cannot give informed consent themselves or by proxy cannot be used in experiments. However, the opposite is common practice: if there are no animal-free alternatives to animal experiments and the scientific or societal advantage is assessed by the Animal Ethics Committee (AEC) as more important than the disadvantage for the animals, no consent is asked from the animals [17]. The AEC assesses the legitimacy of animal use, based on an inventory of the expected scientific and societal benefits and the expected burdens for a specific number of animals from specified species [28]. If the animals are owned by others than the research institute (e.g., pet owners or farmers), additional permission is needed from the owners, based on their informed consent [29].

For many years, I have been taking the role of ethics expert in several Dutch AECs that make an independent ethical assessment of a project proposal including animal experiments. In the Netherlands, the AECs cannot decide on the legal permissibility of the project. They send their advice to the Central Committee (CCD) who will grant or refuse a project license. AECs are invited to not only take the welfare of the animals into account, based on their intrinsic value, but also their physical and genetic integrity. The AEC assessment poses strong dilemmas. Despite the rapid emergence and development of Non-Animal Methods (NAMs), many types of animal research are still not replaceable. This means that trade-offs are made, for example between the chance of finding new treatments that can help numerous patients with severe conditions, developing vaccines, or testing the safety of chemical substances, on the one hand, and the use, suffering, and death of mostly large numbers of animals, on the other hand. This means that they balance the benefit of the experiment against the suffering of the animal. We will not now go into these dilemmas as such but will focus on the possibility to approach them as a human proxy of research animals.

For the sake of a thought experiment and to exclude issues of the translatability of animal research to humans, let us assume that a being has extra reasons to consent to involvement in experiments if they are conducted for the sake of the health and wellbeing of their own species. This is at least what we assume about humans: they are often willing to undergo relatively safe experiments aimed at the health and wellbeing of other humans. This may be comparable to how non-human animals would look at their involvement if they could understand the procedures and risks and benefits. Just like we cannot do without human volunteers for human-aimed medical research, we cannot do completely without animals for veterinary research, at least not yet. We should try to find out what Miffy would do in case of a request to participate in an experiment for the benefit of other rabbits.

Rabbits are social animals. They like to lie and play together. They are very sensitive to some horrifying and often deadly contagious diseases. If we needed a hundred rabbits to find a vaccine against one of those diseases, would these rabbits volunteer? To be able to assess that for them, we would need some extra information. How common is this disease? How grave or lethal is it? What are the odds that a vaccine will be found? How many rabbits will benefit from it? Will it be used to help free-living rabbits? Will it be used to make intensive rabbit farming more profitable? Furthermore, what is expected of the subjects? Will the administration of a substance, observation, and blood drawing be enough? Is there a risk of becoming very ill or dying? Where will the participants live? Under what type of conditions? Will they be killed at the end of the experiment?

Those who are involved in AECs or other ethical bodies will recognise this type of question. It is the same type that is used for ethical assessments. In fact, the questions function as a type of proxy, representing something like ‘society’ or ‘general interest’. But I wonder if they ever ask themselves the following: what would Miffy do if she herself could decide? Would she be prepared to give her life? If not, should we respect that? Would it matter to her if the research is for the benefit of members of her own species or for other sentient beings? Can we imagine, as proxies, that if she knew that other individuals can suffer the way she can, she would feel for them too? Would that change her position? Would she be more inclined to endure some hours of pain and give part of her life if she would have an extra good life in advance and a quick death?

Maybe, if Miffy would be granted a good social life, if the risk would be low, the experiment short, the procedures not too frequent, the life after the experiment long and peaceful, the benefit large for many rabbits or other sentient animals, she would say yes. But would she want to undergo painful experiments and give her life prematurely for this purpose? Would she even want to come into existence? (At the moment of the ethical assessment, the rabbits for the experiments have probably not even been bred yet.) The answer would probably be no. Let us not forget that most laboratory animals are killed as part of the experiment for further research on their organs. Others are killed because they are redundant. Only a very small percentage of the animals stay alive and are rehomed [30]. If the research is conducted on livestock farms, the animals still go to the slaughterhouse to be killed. Nevertheless, to make things more complicated, we should also take into account—by proxy—the welfare, suffering, and death of the other stakeholders, in this case the future rabbits who could be vaccinated if a vaccine were found against a specific rabbit disease.

### 4.3. Broader Applications

Although the application of informed consent by proxy is the most promising of all concepts, it is clear that it is not without limitations. Nevertheless, exploring it further and applying it to specific cases can shed an interesting light on ethical assessments. What if we would apply it to sentient animals used for food (meat, fish, eggs, milk, and dairy products)? It will be immediately clear that animals would not consent to being confined, transported, and killed in an industrial way for that purpose. Maybe a mother mammal will allow for a trusted caretaker to take some of her milk, but not all. She would probably not consent to removing her young from her presence and care, because of her strong caring instincts. Even if a cow would get used to not having her calf with her, the moment of separation is probably deeply traumatic according to the behaviour cows express: howling, attacking farmers, and running after the truck that takes the calf away. Nussbaum puts it like this [10] (p. 165):

“All this is very far from what happens when we eat animals. (I’ll use the food case as my central example, but much animal experimentation has the same problem.) Here we are usually not making a proxy judgement on behalf of the animal, we are pleasing ourselves and using the animal as means to our own ends. I have asked the reader to reject without further argument the factory farming industry, which gives all the animals involved painful and cramped lives without the exercise of characteristic life-activities, such as free movement, social relations, fresh air, and choices of how to spend one’s day. Let’s consider the best cases of humane farming, where the animal has a reasonably good life: good food, fresh air, the companionship of other animals, etc.–and then is killed in a genuinely painless manner.

Some animals we kill for food are very young animals, who have had no chance to unfold their characteristic form of life in mature functioning. [...] It is reasonable to suppose that young creatures learn to seek maturity as a central goal. When that is not reached, they do suffer a grievous type of interruption.”

Ethical assessors, but also animal researchers and other people who use animals, are invited to look at animal use from this new perspective. This would lead to a stricter selection of accepted animal use. What would Miffy do if we would ask her the following: can I kill you and eat your flesh in a nice meal with my family? I think she would refuse. What would a dog decide if I ask him the following: will you come and live with me, so that I can feed, clean, and cuddle you every day? I will walk you three times a day, but you will sometimes have to be on a leash. When you become ill, I will take you to the vet. Maybe he would say yes, most certainly if the alternative were roaming through the fields looking for scraps of food and being in danger of ending as road kill or suffering from an untreated disease. However, there may be dogs, living in a relatively warm climate, who would choose to live freely.

Living or working close to animals, observing them, and communicating with them can help humans to understand them better [31] and thus enhance the quality of their representation as a proxy. A form of thinking about informed consent by proxy was explored in a mock assembly with humans representing cows and future human generations, supported by immersive theatre, storytelling methods, and a meet-up with cows. The participants experienced the mock assembly as potentially able to serve as an intermediary towards interspecies democratic decision making [32]. Although the mock assembly provided an interesting exploration of possibilities, much research is still needed to arrive at a well-functioning and fair system for representing animals in complex decisions.

Sometimes, we will have to make decisions on the level of a group of animals or even a species, answering questions like the following: should it be allowable to keep salamanders in a small area with a pool for the sake of protecting their species? In that case, knowledge of the particular species (e.g., about amphibian migration) combined with imagining and empathizing with the life of the members of the species (what would Sally the salamander herself decide in this case?) would be the best approach. For now, I would like to propose to add the perspective of informed consent by proxy to existing ethical assessment situations.

## 5. Conclusions

We considered assent, alertness to dissent, and informed consent by proxy as approaches for the ethical assessment of activities involving sentient non-human animals. Informed consent by proxy seems to be the best option. Independency, closeness, and expertise are important characteristics of proxies. The assessment should no longer be the following: do the benefits outweigh the harms? Instead, it should be this: what would this animal do with full understanding and knowledge of the request and its implications? Nevertheless, this would also hold for other stakeholders. The assessors would also have to think as proxies about the interests of the patients, for example, or invite real patients to take part in the assessment. This means that there will still be a harm–benefit analysis but with a stronger emphasis on what every stakeholder would want if fully informed.

Proxy assessments will in part remain speculation, but we should do the best we can. Balancing the risks of anthropocentrism and being in anthropodenial, we should find the right place on the scale. In individual cases, we should combine our knowledge of the species and of the individual animal with alertness to the animal’s signals. The animals of most species will hide pain or weakness from humans as a stress-induced reaction. The best way to find out about their pain is to get to know them, as a veterinarian would do when approaching a patient [18]. Other helpful elements are the triangulation of the claims of various experts and, if possible, ongoing bodily assent shown by the animal.

There are of course limitations to this approach. We will never be able to say, as a proxy for animals as we do for humans, that this was my grandmother, she hated the idea of someone touching her dead body, and she was quite stubborn, so I am sure she would never have wanted anyone to take her organs. Nor can we say that my father was a scientist himself: he would have loved the idea of his body being useful for science. Neither will we be able to know if an individual non-human animal would be prepared to risk their life for a greater cause, like some humans do in the armed forces or the fire brigade. And what about future generations of sentient beings? Would non-human animals take them into account? This is very hard to assess.

Additionally, the context of everything discussed in the paper is a human-dominated society, including power-related knowledge systems. In the ideal world, at least the animals of domesticated species should have the right to co-compose society and its norms and institutions together with humans (e.g., microboards of friends), based on their own agency. This is important, because we as proxies have limited knowledge about the diversity of non-human animals and will always be biased, and we tend to make those we represent unheard [33].

There are still many pitfalls to this new approach, making it quite possible that the parties required to conduct an ethical assessment will not agree on ‘what Miffy would do’. Nevertheless, while we strive towards a co-designed society in which free animals can participate as co-citizens, we humans should start experimenting with being the best proxies we can. We can try to learn other species’ languages, with or without the help of artificial intelligence. We will need a lot of practice and research into animal preferences to become better at it. However, in more evident cases, we can already assume what Miffy wants and act upon it.

## Data Availability

No data involved.

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
