# Peer review of "What Would Miffy Do? Applying Informed Consent by Proxy to All Sentient Animals"

_animals, 2024, doi:10.3390/ani14182643_

Round 1
Reviewer 1 Report
Comments and Suggestions for Authors
Well done on covering an increasingly important topic. The study presents an interesting approach to the topic. There are a few minor typographical errors that I have listed in the attached Word document that require attention.
In addition I have a few comments that may be useful or improve clarity of some of the arguments in the paper:
1. In the introduction, the authors makes reference to Kantian Philosophy as the basis for requiring consent from humans. It would perhaps enhance balance, if non-human animals were also looked at through the Kantian lens. Are non-human animals moral participants in society? Kantian Philosophy has been used to argue that non-human animals are not moral agents and therefore do not have inherent legal standing other than as property belonging to moral agents.
2. As the author acknowledges that evidence of agency in non-humans is limited, please clearly state the scope of non-humans under consideration in this paper. Is it all sentients, just those where there is good evidence of agency, all domesticates or a different criteria?
3. The arguments presented in Lines 255-270 (Nussbaum) for representation have a heavy anthropomorphic bias. It is unlikely that cats would pass off the chance to breed because they are concerned about the fate of the kittens. They are more likely driven instinctually to increase the proportion of their genes in the population and the mechanism is unlikely to be conscious.
4. The same concern exists for the the thought experiment in Line 276. The answer to "What would Miffy do?", would realistically be "choose freedom or non-interference". The suggested answer is anthropomorphic.
5. The arguments from Line 304 could be used for humans and yet we seek individual consent as we acknowledge that individual agency matters. Is the argument that this is not so for non-humans?
6. The argument in Line 385 is flowed as it is unrealistic to assume that a cat would choose to have a reflective collar in acknowledgement of the interest of birds and mice. This is an anthropomorphic view and may not respect the primacy of the biological drives of animals.
7. The conclusion in Line 407 does not appear supported by the presented arguments. There has been no critical review in the paper of the merits or otherwise of the current consideration "Do the benefits outweigh the harms" if the principle is applied with equal consideration of equal interest. Before rejecting the status quo we must give it a fair evaluation. The biggest flaw is probably the current implementation of the Utilitarian calculus that is anthropocentric. However, replacement with a potentially anthropomorphic system would be retrogressive.
8. In human law, the use of proxies (agents) relies heavily on the agent being able to represent the true will of the client/ patient. This paper does not offer a clear way of testing this for non-human animals. The current system already has veterinarians, animal keepers as well as lay members in the Animal Ethics Committee. Are these therefore best placed to make representations about the animal's likely consent? One may argue that it is easier to objectively illustrate the harms vs good of the study than to speculate on how the animal would have chosen, had it been fully informed. The examples given, whether they are about humans volunteering (most humans are not volunteering for medical research), about Nussbaum's cat (which driven by biological imperative is likely to choose to continue breeding) and Miffy (who would be driven by instinct to choose freedom unless trained otherwise), all highlight how the proposed approach is likely to lead to arguments.
Overall, I feel the moral status of animals, in particular agency needs to be explored further to build a stronger rationale for the conclusion. As long as non-human animals are viewed as property, decisions will always be made for the greater (human) good than for individual non-human sentients.
A very interesting read.
Issues
1. Lines 14 & 23: [Typo]: ….what they want en to estimate….
2. Line 134: [Typo]: ……They can told, but will probably
3. Line 177: [Spelling]:…… It is important thought that it is exercised
4. Line 217: [Spelling]: ….. If offers the proxy
5. Line 300: {Typo]: ….We will now not go into these dilemma’s a such, but
Comments on the Quality of English LanguageThe paper is clearly written in accessible English. There are a few typographical errors I have highlighted in the attached document, otherwise very good quality of presentation.
Author Response
- In the introduction, the authors makes reference to Kantian Philosophy as the basis for requiring consent from humans. It would perhaps enhance balance, if non-human animals were also looked at through the Kantian lens. Are non-human animals moral participants in society? Kantian Philosophy has been used to argue that non-human animals are not moral agents and therefore do not have inherent legal standing other than as property belonging to moral agents.
Thank you for pointing this out. I see your point. I did not want to go into the several approaches that grant moral standing to sentient animals, because it prefer to take it as a premise. I now made that premise explicit and added even more balance by mentioning several ethical approaches with references (FYI: Singer, Regan, Korsgaard, and Nussbaum). This I added at the end of the introduction:
In this paper, I take as a premise that sentient animals have moral standing and are active subjects of their own lives, based on the work of philosophers of several ethical approaches, such as utilitarianism [7], deontology [8,9], and virtue-based capabilities approach [10].
- As the author acknowledges that evidence of agency in non-humans is limited, please clearly state the scope of non-humans under consideration in this paper. Is it all sentients, just those where there is good evidence of agency, all domesticates or a different criteria?\
I agree that this is an important point. About agency: what I discuss is specifically moral agency, of which the evidence is even more limited. Therefore I am very careful about what animals would do for others if they knew the consequences. Agency as such is acknowledged for at least sentient animals. Therefore, I now added the adjective “sentient” in many places in the text. The demarcation line in the paper is explicitly: sentience.
- The arguments presented in Lines 255-270 (Nussbaum) for representation have a heavy anthropomorphic bias. It is unlikely that cats would pass off the chance to breed because they are concerned about the fate of the kittens. They are more likely driven instinctually to increase the proportion of their genes in the population and the mechanism is unlikely to be conscious.
I see what you mean. For the sake of my argument, I cannot make any changes here. The whole idea of informed consent by proxy is not to merely follow the instinct of the animal (which would probably lead to running away from the vet), but to try to help the animal make choices as if it could oversee the implications of its actions beyond instinct. I see this in terms of De Waal 1999 (reference 27 in the paper) as useful animalcentric anthropomorphism, and not unscientific anthropocentric anthropomorphism. At the same time, I can imagine that this is a critical point for the reader. Therefore I added this at the end of the discussion section:
Critics might raise the objection that thinking and choosing on behalf of the animal tends towards anthropomorphism: ascribing human mental experiences to animals. Yet, the fear for anthropomorphism has been overstated. Just as there is the risk of anthropomorphism, there is also the risk of anthropodenial: underestimating the similarities between humans and non-human animals [27]. This means that the challenge lies in empathizing as a proxy with the animal in such a way that both the distinctive characteristics of the species and the comprehensible similarities to humans, as well as the individual animal's personality, are properly represented.
- The same concern exists for the thought experiment in Line 276. The answer to "What would Miffy do?", would realistically be "choose freedom or non-interference". The suggested answer is anthropomorphic.
This issue is tackled by the above solution, based on the idea that animals are capable of more that we think. This is also supported by former references in the paper, for instance to indications of altruistic behaviour of mice and rats. Nevertheless, I do want to be prudent about that. To the conclusion I added the following statement:
Proxy-assessments will in part remain speculation, but we should do the best we can. Balancing between the risk of anthropocentrism and anthropodenial, we should find the right place on the scale.
- The arguments from Line 304 could be used for humans and yet we seek individual consent as we acknowledge that individual agency matters. Is the argument that this is not so for non-humans?
Around Line 304 it says: Fore the sake of a though experiment and to exclude issues of translatability of animal research to humans, let us assume that a being has extra reasons to consent in involvement in experiments if they are conducted for the sake of the health and wellbeing of their own species.
Yes, the argument is that consent is not asked from individual non-humans agents, while it should be. A human has extra reasons to consent in involvement in experiments for the sake of humans. It is an assumption that I have to make. We can think ‘would they consent?’ but in many cases we can ask them. My point is that we do not ask animals and that we should. I really hope this is clear enough in the paper. If not, please let me know.
- The argument in Line 385 is flowed as it is unrealistic to assume that a cat would choose to have a reflective collar in acknowledgement of the interest of birds and mice. This is an anthropomorphic view and may not respect the primacy of the biological drives of animals.
For this point, I would like once more to refer to my answer to question 3. Altruism has been identified in many species and individuals already, and more is to be expected. Therefore, I do not agree with you. To me, it is imaginable that a cat who would know what their biological drive does to others, knowing there is plenty of food with their human at home, would choose differently than to go on chasing. Nevertheless, this is so hypothetical, that I decided to leave this out and not loose the reader.
And let’s not forget to ask consent by proxy to all relevant stakeholders. This means that if we ask a cat if she wants to roam freely during the nights, we should also think about the hypothetical consent of the birds and the mice she will kill during that night. As a final remark to this case: of course the best option is always to find a solution that strikes a balance without harming anyone, which could be a colorful cat collar.
- The conclusion in Line 407 does not appear supported by the presented arguments. There has been no critical review in the paper of the merits or otherwise of the current consideration "Do the benefits outweigh the harms" if the principle is applied with equal consideration of equal interest. Before rejecting the status quo we must give it a fair evaluation. The biggest flaw is probably the current implementation of the Utilitarian calculus that is anthropocentric. However, replacement with a potentially anthropomorphic system would be retrogressive.
Thank you for noticing that the conclusion goes too far here. It is clear that we do not agree on the impact of anthropomorphism. Hopefully you agree to disagree. Nevertheless, you helped mee see the failure of my argument about the birds and the cat collar. I already left this part out, and now the argument works better.
- In human law, the use of proxies (agents) relies heavily on the agent being able to represent the true will of the client/ patient. This paper does not offer a clear way of testing this for non-human animals. The current system already has veterinarians, animal keepers as well as lay members in the Animal Ethics Committee. Are these therefore best placed to make representations about the animal's likely consent? One may argue that it is easier to objectively illustrate the harms vs good of the study than to speculate on how the animal would have chosen, had it been fully informed. The examples given, whether they are about humans volunteering (most humans are not volunteering for medical research), about Nussbaum's cat (which driven by biological imperative is likely to choose to continue breeding) and Miffy (who would be driven by instinct to choose freedom unless trained otherwise), all highlight how the proposed approach is likely to lead to arguments.
I think you have a valid point here. Yet, I think I have offered a new way of looking at the position of the animal in the ethical assessment of animal experimenting by those you mention. I think it is more a matter of opening up new perspectives than starting concrete new routines.
I added at the end of the Discussion section:
For now, I would like to propose to add the perspective of informed consent by proxy to existing ethical assessment situations.
And I added to the limitations in the Conclusion section the following sentence:
There are still many pitfalls to this new approach, making it quite possible that parties required to conduct an ethical assessment will not agree on ‘what Miffy would do’.
Overall, I feel the moral status of animals, in particular agency needs to be explored further to build a stronger rationale for the conclusion. As long as non-human animals are viewed as property, decisions will always be made for the greater (human) good than for individual non-human sentients.
Thank you for this final statement, which is quite realistic. Hopefully the nuance in the last sentence above is helpful.

Reviewer 2 Report
Comments and Suggestions for Authors
Thank you for submitting your article,
I think it would be interesting to critique some of the models that you cite (e.g. reference 28 - the majority of participants identified as vegetarian or vegan, I believe they did not identify their religions - how would this translate to a citizen's assembly representative of the general population?)
Additionally, there are examples where humans choose to sacrifice their lives - for example by volunteering to join armed forces - could this be a worthwhile comparison for considering the motivations of animals in terminal experiments?
Is there evidence that present humans give appropriate moral consideration to future populations? If not, would we anticipate animals giving any consideration to potential future populations?
Comments on the Quality of English LanguageThere are a number of typographic errors to address.
Author Response
- I think it would be interesting to critique some of the models that you cite (e.g. reference 28 - the majority of participants identified as vegetarian or vegan, I believe they did not identify their religions - how would this translate to a citizen's assembly representative of the general population?)
Thank you for this question. I am really trying to be descriptive here, citing what I read In the report. I think it is a very good point that het participants were a very specific group, but to me it was just an example of people trying to do something that happens to be somewhat in line with my argument. Although I cannot offer a full solution here (that would call for an extra paper), I added a critical note:
Although the mock assembly provides an interesting exploration of possibilities, much research is still needed to arrive at a well-functioning and fair system for representing animals in complex decisions.
- Additionally, there are examples where humans choose to sacrifice their lives - for example by volunteering to join armed forces - could this be a worthwhile comparison for considering the motivations of animals in terminal experiments?
Thank you for this important point. You are right that I did not make this explicit. I think it will be very difficult to asses. I decided to add this to the limitations in the last section:
Neither will we be able to know if an individual non-human animal would be prepared to risk their life for a greater cause, like some humans do in the armed forces or the fire brigade.
- Is there evidence that present humans give appropriate moral consideration to future populations? If not, would we anticipate animals giving any consideration to potential future populations?
Thank you for pointing this out. It is an extra topic, that is – in my opinion – too large to cover exhaustively in this paper. But I do want to acknowledge its importance. Therefore, I put the issue forward as a question, also in the limitations section:
And what about future generations of sentient beings? Would non-human animals take them into account? This is very hard to assess.
